# Is There Scope for a Novel Mycelium Category of Proteins alongside Animals and Plants?

**DOI:** 10.3390/foods9091151

**Published:** 2020-08-21

**Authors:** Emma J. Derbyshire

**Affiliations:** Nutritional Insight, Surrey KT17 2AA, UK; emma@nutritional-insight.co.uk

**Keywords:** fungal protein, health, human research, mycelium, mycoprotein, protein classification, scientific awareness

## Abstract

In the 21st century, we face a troubling trilemma of expanding populations, planetary and public wellbeing. Given this, shifts from animal to plant food protein are gaining momentum and are an important part of reducing carbon emissions and consumptive water use. However, as this fast-pace of change sets in and begins to firmly embed itself within food-based dietary guidelines (FBDG) and food policies we must raise an important question—is now an opportunistic time to include other novel, nutritious and sustainable proteins within FBGD? The current paper describes how food proteins are typically categorised within FBDG and discusses how these could further evolve. Presently, food proteins tend to fall under the umbrella of being ‘animal-derived’ or ‘plant-based’ whilst other valuable proteins i.e., fungal-derived appear to be comparatively overlooked. A PubMed search of systematic reviews and meta-analytical studies published over the last 5 years shows an established body of evidence for animal-derived proteins (although some findings were less favourable), plant-based proteins and an expanding body of science for mycelium/fungal-derived proteins. Given this, along with elevated demands for alternative proteins there appears to be scope to introduce a ‘third’ protein category when compiling FBDG. This could fall under the potential heading of ‘fungal’ protein, with scope to include mycelium such as mycoprotein within this, for which the evidence-base is accruing.

## 1. Introduction

Population growth, climate change, food ethics and health are intricately intertwined—higher population growth entails more food production, more emissions and greater vulnerability to climate-related and health impacts [1]. Given this, viewpoints on what constitutes an ‘ideal diet’ are changing rapidly. The EAT-Lancet Food in the Anthropocene article published in January 2019 has been a powerful game changer—highlighting how food systems must urgently transform to feed growing populations in a manner that both supports environmental sustainability and human health [2].

Presently, there are sizeable discrepancies between protein guidance across global food-based dietary guidelines (FBDG). A review paper [3] evaluating FBDG across 90 countries demonstrated how interchangeable these are—countries display and describe what can generally be termed ‘protein foods‘ in a wide variety of ways. Seventy-four percent of FBDG included a key message relating to protein foods, which included fish (58% of countries), meat (53%), legumes (41%), eggs (31%), poultry (29%), dairy (9%), nuts/seeds (8%), and insects (only Kenya) [3]. Half of all countries with food key messages encompassing protein included animal and plant sources, whilst 11% did not have a protein foods key message at all [3]. Within the global FBDG analysis alternative sources of protein, such as fungal protein, were not mentioned.

Fungi were once considered to be plants due to them growing in soil and having rigid cell walls, but are now independently placed in their own kingdom as they are biologically distinct from plant-and animal-sourced foods [4]. The mycelium is the vegetative part of the fungi, but unlike yeast cells, which typically grow as a single cell, mycelium is filamentous and comprised of a mass of branching hyphae [5]. In nature, the mycelium plays a vital role within ecosystems by forming symbioses with trees and recycling carbon—it is interconnected within the soil with such complexity that some authors have coined the phrase ‘wood wide web’ [6]. Mycoprotein is one example of an important and well-established whole-food mycelium protein [7]. Derived from the filamentous fungus *Fusarium venenatum,* this was first discovered in the 1960s [8]. Since then, mycoprotein has been successfully produced by fermentation processes over the last three decades (please refer to Finnigan et al., 2011 for methods) [9]. Mycoprotein also has a benign environmental footprint, which has been demonstrated by lifecycle analysis [10].

Increasingly, consumers are becoming interested in meat analogs (imitators of meat), namely due to accelerating health and environmental awareness [11,12]. Subsequently, simulated meat products can provide high biological protein to consumers without placing heavy burdens on global natural resources [13]. Subsequently, meat analogs which do not involve raising livestock are predicted to expand within food markets, particularly in Asian countries where there are growing demands for these [11]. Mycelium proteins show great promise in achieving this, yet when compared with plant-based proteins they have been comparatively overlooked.

Given continuing changes to contemporary dietary patterns, the present paper evaluates how food proteins are presently categorised within FBDG and discusses whether there is potential to introduce other categories, including mycelium/fungal protein within these.

## 2. Methods

A PubMed was search was undertaken to obtain English-language, peer-reviewed systematic or meta-analytical papers published during the last 5 years. The search terms “animal-protein”, “plant-protein” and “fungal-protein/mycelium/mycoprotein/mushrooms” and “health” were applied. Publications were excluded if they:(1)Did not focus on human, adult populations;(2)Were not a systematic review or meta-analysis publication;(3)Focused on one specific global population/region;(4)Did not clearly specify the protein of focus;(5)Had research outcomes unrelated to health.

Some publications overlapped i.e., included both animal and plant-derived protein sources thus were placed under the category where significant findings were observed. A total of 15, 68 and 8 articles were identified for animal-derived, plant-derived and fungal-derived sectors, respectively. After screening for relevance 4, 9 and 4 publications were identified from each of the specified sectors. These findings are these are summarized in Table 1.

## 3. Animal-Derived Proteins

Meat production has been intensifying since the 1960s, but even more so from the 1980s to the present day, with some suggesting that the supply of meat products has risen by as much as 204% [31,32]. Over time there has been progressively more criticism over animal-derived proteins, particularly red and processed meats and the role that these have to play in the aetiology of chronic diseases and preterm mortality, although extended factors such as cooking methods also need to be better considered [33,34].

Animal-derived proteins are useful sources of bioavailable amino acids and micronutrients [35,36,37]. Different types of meat and fish, however, have been associated with different health outcomes [38]. For example, as shown in Table 1 meta-analytical data shows that substituting red and processed meats with white meat could reduce stroke risk [16]. Increasingly, red and processed meats have, however, been associated with elevated type 2 diabetes (T2D) risk, whilst egg and fish intakes do not appear to be as strongly related [15,17].

In the United Kingdom, the Department for Environment, Food and Rural Affairs and Food Standards Agency define ‘meat’ and other similar specific terms like ‘beef’, ‘lamb’ and ‘chicken’ as mammal or bird skeletal muscle with natural tissue that is fit for human consumption [39].

The World Health Organisation has defined red meat as: “all mammalian muscle meat, including, beef, veal, pork, lamb, mutton, horse, and goat” [40]. Equally, processed meat has been defined as: “Meat that has been transformed through salting, curing, fermentation, smoking, or other processes to enhance flavour or improve preservation. Most processed meats contain pork or beef, but processed meats may also contain other red meats, poultry, offal, or meat by-products such as blood. Examples of processed meat include hot dogs (frankfurters), ham, sausages, corned beef, and biltong or beef jerky as well as canned meat and meat-based preparations and sauces” [40]. Even though these definitions have been clearly specified, the use and application of these vary extensively within epidemiological and observational research. For instance, a recent review pooling data from 369 articles evaluating ‘muscle food consumption’ highlighted that there were 1020 different dietary meat categories and 776 inter-changeable descriptions [41].

Another escalating concern of existing relevance is rising numbers of outbreaks of animal diseases, including avian influenza which are contributing to mounting distrust amongst consumers [42]. It has also been proposed that wild meat trade markets should be urgently disbanded given that these have been related to viral infections [31].

## 4. Plant-Derived Proteins

Definitions of ‘plant-based’ diets are being using widely within scientific and social domains. A basic internet search of ‘definition of plant-based diets’ yields around 82,900,000 results, many of which are somewhat nebulous. Earlier work discusses how plant-based foods have been common staples of traditional diets in Mediterranean and Asian regions, which typically include the consumption of large proportions of fruit, vegetables, legumes, whole grains, and nuts and smaller amounts of refined grains and red meat [43]. The EAT-Lancet Planetary Health Plate is also described as being predominantly ‘plant-based’ and broadly defines this as being comprised of approximately 50% or 500 g of fruit and vegetables daily [44].

Within the large US National Health and Nutrition Examination Survey (NHANES) an analysis of ‘plant-based’ foods defined these as having zero servings of dairy, meat, poultry, fish and eggs but providing servings of fruits, vegetables, legumes, grains, soy products and nuts/seeds [45]. The additional category of ‘higher protein plant-based foods’ was also used and comprised beans, peas, legumes, nuts, seeds and processed soy products [45]. Subsequently within the umbrella term ‘plant proteins’ there are many different food derivatives. For example, soybeans, chickpeas, soft wheat, alfalfa and spirulin are all examples of plant proteins, as are pulses such as peas, faba beans and lupins, although these tend to be used less, possibly due to less favourable sensory perceptions [46]. Quinoa and hempseed are other examples of plant proteins that are well regarded for their fibre, micronutrient and polyphenol profiles [47].

The terms ‘vegetarian’ and ‘vegan’ have had proposed legal definitions put forward by the German Federation for Food Law and Food Science, which have been looked on favourably by the German federal state [48]. Unfortunately, no such formalisation exists for the definition of ‘plant-based’. Thus, quite often vegetarian, vegan, flexitarian and other deviations of different diets are being categorised as being plant-based. It is believed that the European Commission is also looking into the process of establishing legal definitions for vegetarian and vegan foods, but given the rapid state of changes towards plant-based diets, there is an urgent need to consider a formalised definition for this too [49].

As shown in Table 1, plant-based diets are increasingly being linked to health and well-being. For example, in the last five years alone five synthetic reviews have observed benefits for cardiovascular disease mortality, improved lipid profiles, markers of inflammation and glycaemic control, especially amongst participants with poor health i.e., hypercholesterolemia at baseline [18,19,24,25,26]. Of the plant-based proteins, the evidence for soy protein appears to be particularly strong, with potential benefits for blood pressure and low-density lipoprotein reduction [20,23] and gains in strength and lean body mass [21].

## 5. Mycelium/Fungal-Derived Proteins

Fungi have been categorised as a separate and indeed ‘Third Kingdom’ due to their distinct cellular organisation, with these falling outside the dichotomy of animals and vegetables [50,51]. Previously Feeney and colleagues [4] have listed the major differences that separate fungi from animals and plants explaining that: (1) plants produce food through photosynthesis and have chlorophyll, (2) animals ingest their food and (3) fungi lack chlorophyll, exist on decaying material or can be grown using various substrates commercially. Fungi also contain chitin (a polysaccharide derivative of glucose) rather than cellulose, which is found in plants and ergosterol instead of cholesterol, which is found in animal and mammalian cells [4]. Their cell walls are a useful provider of beta glucan (1,3 and 1,6 linkage) [52].

Globally, fungal-derived proteins been gaining popularity. For example, global mushroom and truffle production has been reported to have grown from 500% between 1980 and 2011 [53]. The main five mushroom producers in the world in 2011 were reported to be The Netherlands, Poland, Italy, China and the United States [54]. In terms of mushroom and truffle global production China has produced 65% since 2009, the European Union (EU) 24%, the United States 5% and Canada, Indonesia and Japan about 1% each [54]. Mycoprotein is a widely accepted food product—sold as Quorn^TM^. It is consumed internationally across 17 countries which includes the United States [55]. In the United Kingdom (UK) in 2019 Quorn^TM^ was positioned as the meat-free market leader and 39th largest UK brand with sales equivalent to £188.3 million and £12.9 million invested in New Product Development [56]. Quorn^TM^ food shortages were observed early in 2020 in the UK due to a surge in popularity during Veganuary [56].

### 5.1. Mushrooms

It is well established that mushrooms are fungi—key planetary components that are often referred to as the “Forgotten Kingdom” [4,57]. These have been eaten over centuries with ancient Greeks claiming that mushrooms provided warriors in battle with strength, and similarly, Romans viewed them as the “Food of the Gods” [58]. Mushrooms are rarely discussed as a protein source for archaeological populations yet have relatively high nitrogen isotope values that could mimic meat-eating [59].

Nutritionally mushrooms typically comprise 50% to 65% total carbohydrate, 19% to 35% proteins (including lectins with proposed biological and medicinal activities), 2% to 6% fat (unsaturated fatty acids predominate over saturated fatty acids), are abundant in fat soluble vitamins and, along with their ergosterol content, are believed to be the only vegetarian source for vitamin D [60,61]. Alongside this, mushrooms also provide fibre polysaccharides and a range of bioactive molecules including β-glucans, triterpenoids and antioxidants which are gaining interest for their biological and medicinal properties [60]. The nutritional profile of different mushroom forms is summarised in Table 2, with dried mushrooms clearly concentrating nutritional values.

From a health stance the Mushrooms and Health Summit Proceedings suggested some benefits for cognition, breast cancer risk reduction, weight management and oral health [54]. Meta-analytical work also shows that mushroom intake could be inversely associated with cancer risk, particularly breast cancer [63,64,65]. It has been also been proposed that mushroom consumption could protect against obesity-related hypertension and dyslipidaemia, possibly by mediating antioxidant and anti-inflammatory pathways [66,67]. As well as possessing antioxidative, anti-inflammatory and immunomodulating activities, it is also thought that mushroom polysaccharides could act as prebiotics in the digestive system [68].

More research is needed to study habitual intakes of mushrooms, and indeed, other fungal protein sources in relation to specific markers of health, as observational studies tend to overlook this important dietary constituent. One study, however, investigated the effects of eating 226 g of roasted *Agaricus bisporus* mushrooms finding that this significantly increased average stool weight and resulted in a greater abundance of Bacteroidetes (beneficial gut bacteria) and reduced levels of less favourable Firmicutes [69]. As shown in Table 1 a growing number of synthetic reviews have collated the health effects of mushroom and polysaccharide consumption, with some evidence for improved cardiometabolic health, vitamin D status and immune function [28,29,30]. Ongoing work is now needed to build on these provisional findings.

### 5.2. Mycoprotein

Mycoprotein, which is produced from *Fusarium venenatum,* is also a naturally occurring fungus [9]. Over 50 years ago Lord Rank of the Rank Hovis McDougall group sought to find a solution when there were similar problems to those of present day—population growth and food shortages [8,55]. His team of scientists collected and tested more than 3000 soil organisms until they discovered that *Fusarium venenatum*, a filamentous micro fungus that could fulfill such a need [55]. After a 10-year evaluation process, in 1983 the UK Ministry of Agriculture, Fisheries and Food approved mycoprotein for food use. Since then mycoprotein has been approved for sale in all EU counties, Switzerland and Norway, the USA, Canada, Australia and New Zealand and recently in Thailand [55]. In addition, a growing body of scientific evidence has looked at the metabolic, anabolic and broader health effects associated with mycoprotein consumption [7,8,55].

In its food ingredient form, mycoprotein is relatively low in energy, total saturated and saturated fatty acids and a good provider of dietary fibre (Table 2). When compared and aligned against European Commission nutrition claims mycoprotein may be regarded as being: (1) ‘high in protein’ i.e., at least 20% of the energy value of the food is provided by protein, (2) ‘low in fat’ i.e., contains no more than 3 g of fat per 100 g of solids, (3) ‘low in saturated fat’ i.e., does not contain more than 1.5 g of saturated fatty acids per 100 g of solids and (4) ‘high in fiber’ i.e., contains at least 6 g of fiber per 100 g [70,71]. This fungal protein provides a wide spectrum of inorganic compounds and, when applying European Commission nutrient claims, mycoprotein falls under the category of being a source of riboflavin, folate, vitamin B12, phosphorous, zinc and manganese [70]. It contains 180 mg choline per 100 g which has recently been highlighted as a nutrient of concern that could be lacking from diets in general, but particularly vegetarians and vegans who do not consume animal-derived foods—some of the main dietary providers of choline [72].

Mycoprotein is a good source of quality protein and abundant in essential amino acids (EAA)—providing all nine EAA (Table 3). As a percentage of total protein, its EAA composition is 41%—similar to spirulina, making this higher than most other commonly consumed plant-based protein [55]. The Protein Digestibility Corrected Amino Acid Score (PDCAAS) ratio for mycoprotein has been shown to be 0.996 where measures of digestibility were characterised with human ileostomy patients [73]. The bioavailability of the protein in mycoprotein has been characterised where the effect of mycoprotein ingestion (a test drink providing a mass-matched 20 g, 40 g, 60 g or 80 g bolus of mycoprotein or 20 g milk protein) on EAA concentrations was studied in a single-blind, randomised, crossover study conducted on 12 healthy young males [74]. Mycoprotein ingestion led to slower and more sustained EAA and branched chain amino acid levels compared with milk, with evidence of a plateau at intakes of 60 to 80 g bolus of mycoprotein [74]. In addition, such high bioavailability has now been shown to stimulate muscle protein synthesis rates to a greater extent than milk protein in rested and exercised skeletal muscle of healthy young men [75].

Interestingly, further research has suggested that it is the whole-food nature of the mycelium that is important in delivering this anabolic response [75]—a feature also demonstrated for whole egg where the consumption of whole eggs promotes greater stimulation of post-exercise muscle protein synthesis than the ingestion of isonitrogenous amounts of egg whites in young men [76]. Subsequently, both the bioavailability and the amino acid composition of mycoprotein position make it a promising dietary protein source—having potential to support skeletal muscle protein metabolism [7].

As well as work on protein quality and anabolism, the scientific evidence-base has been building allowing the impact of diets rich in fungal protein mycoprotein on metabolic markers of health to be characterised [8]. Several studies have observed either reductions in energy intakes or satiety effects associated with mycoprotein consumption [77,78,79,80]. A large body of evidence has documented favourable changes in blood lipid levels, including reduced plasma cholesterol, low-density lipoprotein and improved high-density lipoprotein [81,82,83,84,85,86]. Current work has shown that daily mycoprotein consumption over the course of one week modulates the plasma lipidome (the totality of lipids in cells), building on the theory that mycoprotein favourably modulates lipid regulation [87]. Other work has observed notable improvements in markers of glycaemia and insulinaemia, including reduced insulin levels and sustained hyperinsulinaemia and hyperaminoacidaemia [74,77,88,89]. As shown in Table 1, results collated from five trials found that mycoprotein ingestion was associated with reduced insulin levels and appeared effective at reducing ad libitum energy intake in health lean, overweight and obese adults [27].

One school of thought is that these beneficial effects could be attributed to the profile (amount and type) of dietary fibre present in mycoprotein i.e., its chitin and beta-glucan profile [8,90]. Reduced energy intakes associated with fibre consumption could partly be due to the actions of short chain fatty acids (SCFAs) which are yielded from fibre by colonic bacteria [90]. Novel work shows that mycoprotein and extracted mycoprotein fibre are highly fermentable, resulting in a total SCFA production of 24.9 and 61.2 mmol/L respectively [90]. Another possibility is that the phytochemicals/pigments present in fungi could be inducing some of these effects. For example, an extraordinary range of colours are known to be produced by fungi including pigments such as azaphilones, flavins, melanins, phenazines, and quinines [91]. Continued research is needed.

### 5.3. Others

Single-cell proteins (SCPs) that can be produced by fungi are also gaining interest. SCPs grow rapidly and have a high protein content yet have minimal dependence of soil, water and climate conditions [92]. Some work has investigated the production of a vegan-based mycoprotein concentrate using a pea-industry by-product, again using edible filamentous fungi [93]. It was proposed that this protein source could help to provide protein and energy to the one billion people in the world failing to have access to this [93].

## 6. Discussion and Future Directions

Contemporary dietary patterns are changing—a shift being fueled by population growth, environmental awareness and health [94]. Subsequently, food systems, dietary viewpoints and guidelines are transforming. The 2019 EAT-Lancet Commission Food in the Anthropocene report [2] and its reflections on current food production systems has been at the epicentre of a wave of change, which is diffusing.

Already change can be seen in updated 2019 Australian Dietary Guidelines [95] which have embedded core elements from the planetary health diet—making vegetables and fruits half of the plate and providing advice to limit red meat and avoid processed meat, while opting for fish, chicken, beans, nuts and tofu as healthy and versatile protein sources. Whilst such changes are highly commendable in the bid to consume more ‘plant-based’ protein it could be questioned whether other valuable sustainable protein sources are being overlooked.

Fungi have distinct nutritional and bioactive profiles and the evidence-base with regard to health benefits has also been building. This is particularly evident with the mycelium mycoprotein, which has been successfully and sustainably produced since 1985 [55]. Mycoprotein is high in protein, provides the full range of essential amino acids, is low in fat, low in saturated fat, high in fibre and a source of riboflavin, folate, magnesium, phosphorous, copper, zinc, selenium and choline [96]. Alongside the recent systematic review showing that mycoprotein ingestion is associated with reduced insulin levels [27] at least 13 separate human studies have studied inter-relationships between mycoprotein consumption and human health [8], and more in recent years. Fungi are known to be rich sources of ergothioneine and glutathione, which are both potent antioxidants with potential anti-inflammatory properties [97]. The antiaging effects of bioactive molecules isolated from fungi are also beginning to be identified [98]. Similarly, the role of fungal metabolites in the gut ‘mycobiome’ and its relation to the onset of human diseases is also gaining attention [99].

Shifting protein intakes towards plant and fungal-derived proteins could also yield benefits from an environmental point of view. Table 4 provides a sustainability comparator and evaluates the carbon, land and water footprints of different food-based proteins. It can be seen that both plant-based protein and fungal protein have the lowest average carbon footprints (1.02 kg CO_2_e kg^−1^ and 0.8 CO_2_e kg^−1^, respectively). Beef has the highest land footprint of 0.0068 ha kg^−1^) and mycoprotein the lowest at just 0.00018 ha kg^−1^. Regarding water use—green, blue and grey water usage are highest for meat production and lowest for mycoprotein production. Figure 1 shows the global warming potential (GWP) of mycoprotein compared against different plant proteins showing that this is substantially lower than soybean protein isolate.

In terms of marketability, there is a growing demand for alternative and sustainable protein sources with a suitable nutritional profile and pleasant sensory attributes [100]. The public are becoming increasingly aware of the need for green yet cost-effective processing technologies [100] and the trend of flexitarianism has been gaining in popularity, particularly amongst females [101]. Given shifting consumer trends and health evidence, now appears to be the time to re-evaluate FBDG in the context of three distinct and separate food kingdoms—plant, animal and mycelium/fungal [4]. Given the large inconsistencies in protein guidance across global FBDG, fresh approaches need to be taken [3]. This includes placing an emphasis on consuming protein from various ‘sectors’ which could potentially be broadened into: animal-based protein, plant-based protein and mycelium/fungal-based protein.

Alongside this, the definition of ‘plant-based’ should be formally defined, as this is subject to misinterpretation. For example, extruded foods rich in plant protein are being developed to meet the needs of consumers [102]. It is unclear where such products would fit within food models. Fungi, by contrast, are a ‘whole-food source’ that are produced naturally in nature—just like fruits, vegetable- and animal-based protein and innately lends itself to the formation of a new food group.

**Table 4 foods-09-01151-t004:** Sustainability comparator of different protein sources—summary of the average carbon, land and water footprints.

	Carbon (kg CO_2_e kg^−1^)	Land (ha kg^−1^)	Green Water (L kg^−1^)	Blue Water (L kg^−1^)	Grey Water (L kg^−1^)
**Animal-Based**					
Beef, general	16.2	0.0068	13,921	752	1016
Beef, mixed	26.7	0.0035	15,500	250	4000
Beef, grazed	121	0.0049	16,500	300	5000
Pork	8.29	0.0012	5070	416	509
**Plant-Based**					
Soy	1.02	0.0014	1855	240	573
**Fungal**					
Mycoprotein	0.8	0.00018	539	35	202

Source: Carbon Trust [10,103,104].

**Figure 1 foods-09-01151-f001:**
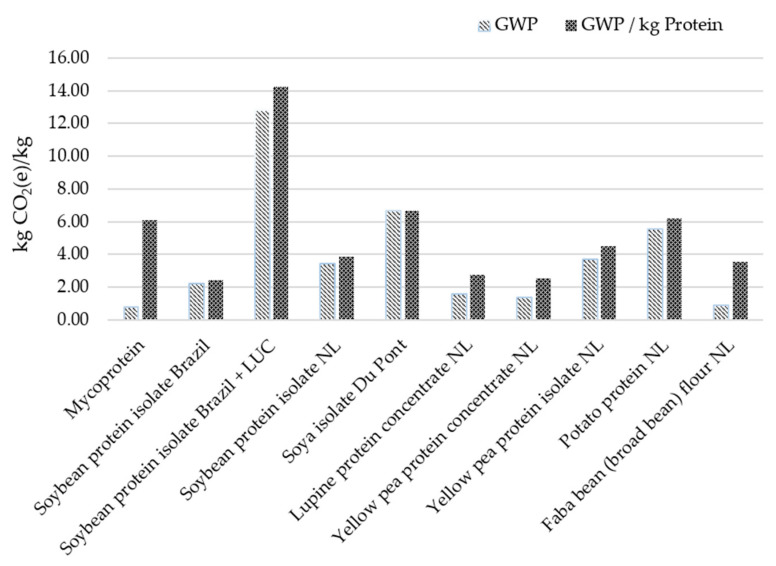
Comparison of mycoprotein global warming potential (GWP) with plant protein. Source: Finnigan et al. (2020) [105].

Taken together, future shifts in protein trends needn’t result in byzantine definitions or group categories. Food policies and dietary guidelines should now be extended beyond going ‘plant-based’ and encompass protein consumption from different and newly specified food groups—one of which could potentially be fungal protein, or mycoprotein.

## 7. Conclusions

Food systems are undoubtedly changing—a change driven by complex interplay between population expansion, sustainability concerns and public health. This is having a domino effect on food policies and dietary guidelines, which are swiftly shifting. Whilst such changes are highly commendable, the current fashion to eat ‘plant-based’ diets could also be extended to other sustainable whole-food proteins. Bearing this in mind, it is now time for mycelium/fungal protein to come out of the shadows and position itself as a pivotal, third category of protein that sits comfortably beside plant-based and animal-based protein.

## Figures and Tables

**Table 1 foods-09-01151-t001:** Key systematic reviews and meta-analysis papers investigating animal, plant or fungal proteins and health.

	Data Collation	Main Findings
**Animal-Derived Proteins**		
Guasch-Ferré et al. (2019) [14]	36 studies	Substituting red meat with high-quality plant protein sources (but not fish or low-quality carbohydrates) resulted in favorable changes to blood lipids and lipoproteins.
Fan et al. (2019) [15]	12 studies	T2D risk ↑ with ↑ consumption of total protein and animal protein, red meat, processed meat, milk, and eggs. Plant protein and yogurt had an inverse relationship.
Kim et al. (2017) [16]	15 studies	Inter-relations between meat intake and stroke risk differ by type of meat. Red and processed meats replaced with white meat may be considered for stroke prevention.
Tian et al. (2017) [17]	11 studies	Red meat and processed meat are risk factors for T2D whilst soy and dairy products appear protective. Egg and fish intake were not associated with decreased T2D risk.
**Plant-Derived Proteins**		
Naghshi et al. (2020) [18]	32 studies	Intake of plant-protein was associated with a lower risk of all-cause and cardiovascular disease mortality.
Zhao et al. (2020) [19]	32 studies	Plant-protein could improve lipid profile in patients with hypercholesterolemia.
Blanco Mejia et al. (2019) [20]	46 trials	Soy protein significantly reduced LDL cholesterol by approximately 3–4% in adults.
Messina et al. (2018) [21]	9 studies	Soy protein supplementation produced similar gains in strength and LBM in response to RET as whey protein.
Shams-White et al. (2018) [22]	7 trials	Soy protein consumption versus animal protein was not more advantageous at improving markers of bone health.
Kou et al. (2017) [23]	12 trials	The ingestion of ≥25 g soy protein per day had BP-lowering effects, possibly due to isoflavones.
Li et al. (2017) [24]	112 trials	Plant protein in substitution for animal protein ↓ LDL cholesterol by 0.16 mmol/L, non-high-density lipoprotein cholesterol by 0.18 mmol/L and apolipoprotein B by 0.05 g/L.
Eichelmann et al. (2016) [25]	29 trials	Plant-based diets were associated with improved markers of inflammation, including CRP and IL-6.
Viguiliouk et al. (2015) [26]	13 trials	Replacing animal with plant protein leas to modest improvements in glycemic control in individuals with diabetes.
**Fungal-Derived Proteins**		
Cherta-Murillo et al. (2020) [27]	Five trials	Results showed that mycoprotein reduced insulin levels. Acute mycoprotein intake also decreased energy intake at an *ad libitum* meal and post-24 h in healthy lean, overweight and obese humans.
Dicks and Ellinger (2020) [28]	Eight trials	*P. ostreatus* intake may improve cardiometabolic health, but ongoing research is needed.
Cashman et al. (2016) [29]	Six trials	Consumption of ultraviolet (UV)-exposed mushrooms may increase serum 25(OH)D when baseline vitamin D status is low.
Fritz et al. (2015) [30]	28 studies	Polysaccharide K may improve immune function, reduce tumor-associated symptoms, and extend survival in lung cancer patients.

Key: BP, blood pressure; CRP, C-reactive protein; IL-6, interleukin-6; LBM, lean body mass; LDL, low-density lipoprotein; RET, resistance exercise training; T2D, type 2 diabetes.

**Table 2 foods-09-01151-t002:** Nutritional profile of fungal proteins.

	Shitake, Dried, Raw	Oyster Mushrooms, Raw	White Mushrooms, Raw	Mycoprotein per 100 g (Wet Weight) *
Energy, kcal	296	8	7	85
Protein, g	9.6	1.6	1.0	11
Total fat, g	1.0	0.2	0.2	2.9
Saturated fatty acids, g	0.2	Tr	0.04	0.7
Monounsaturated fatty acids, g	0.3	Tr	Tr	0.5
Polyunsaturated fatty acids, g	0.1	0.10	0.11	1.8
Total carbohydrate, g	63.9	Tr	0.3	3.0
Sugars, g	N	Tr	0.3	0.5
Dietary fiber, g	N	N	1.2	6.0
Riboflavin, mg	1.27	0.4	0.27	0.26
Folate (B9), μg	N	N	40	114
Vitamin B-12, μg	0.0	0.0	0.0	0.71
Choline, mg	-	-	-	180
Phosphorous, mg	290	120	94	290
Zinc, mg	N	Tr	0.6	7.6
Iron, mg	1.7	1.9	0.21	0.39
Manganese, mg	N	3.6	0.05	4.9
Sodium, mg	13	77	4	5.0

Sources: Finnigan et al. (2019) [55]; McCance and Widdowson’s (2019) [62]. * Updated micronutrient data provided by Marlow foods.

**Table 3 foods-09-01151-t003:** Essential amino acid profile of different protein sources (g amino acids per 100 g).

	Animal-Based Protein		Plant-Based Protein		Mycoprotein
	Beef	Cows Milk	Soy Isolate *	Soy Concentrate *	
Histidine	0.30	0.09	0.6	0.4	0.39
Isoleucine	0.87	0.20	1.1	0.8	0.57
Leucine	2.53	0.32	1.8	1.3	0.95
Lysine	1.60	0.26	1.4	1	0.91
Methionine	0.50	0.08	0.3	0.2	0.23
Phenylalanine	0.76	0.16	1.1	0.9	0.54
Tryptophan	0.22	0.05	0.3	0.2	0.18
Threonine	0.84	0.15	0.8	0.7	0.61
Valine	0.94	0.22	1.1	0.8	0.60

* Soy isolate and concentrate data have been adjusted to the same water content as mycoprotein. Source: Data provided by Marlow foods.

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
