# Peer review of "Is There Scope for a Novel Mycelium Category of Proteins alongside Animals and Plants?"

_foods, 2020, doi:10.3390/foods9091151_

Round 1

Reviewer 1 Report

Report 3

The paper has improved by revision. It seems to be somewhat hastily done as the claimed Table 1 summarising the nutritional profile of 218 different mushroom forms is missing. Instead, the current Table 1 presents the results from systematic reviews and meta-analyses of different protein sources. Could the author add the missing table and check and rearrange the numbering of the tables. There are some typing errors as well, and missing commas.    

Author Response

The paper has improved by revision. It seems to be somewhat hastily done as the claimed Table 1 summarising the nutritional profile of 218 different mushroom forms is missing. I took this out as there were a lot of new tables and charts but have but this back in now and all of the subsequent tables/figures have been re labelled.  I am not sure where 218 has come from as there is no mention of 218 in the paper? The table includes the main mushroom forms only. 

Instead, the current Table 1 presents the results from systematic reviews and meta-analyses of different protein sources. Please see comment above. Could the author add the missing table and check and rearrange the numbering of the tables. There are some typing errors as well, and missing commas.  The paper have been re-edited and extra commas inserted - hopefully this reads smoothly and well now. 

Reviewer 2 Report

The author followed the suggestions and improved the paper considerably. The paper is now well written and interesting for the reader with broad scientific examples of the topic.

Author Response

Many thanks for the useful feedback and I am very happy that you enjoyed the updated paper.

This manuscript is a resubmission of an earlier submission. The following is a list of the peer review reports and author responses from that submission.

Round 1

Reviewer 1 Report

This is an interesting and a very current topic. However, the author should improve the paper, give more information about this thematic, otherwise it's an incipient paper. For example
Line  65-75. Animal-Derived Proteins. The author never writes about "white meat" which is an important source of proteins, as well as fish or eggs.
Line 77-97. Plant-derived Proteins. The author could give examples of different plant protein amounts that contributed to human diet.
Mycelium/Fungal-Based Proteins. Different fungal contained different proteins? Please discuss.
Table 3. Interesting information but the authors could add other data related with plants (for example)

Author Response

This is an interesting and a very current topic. However, the author should improve the paper, give more information about this thematic, otherwise it's an incipient paper. For example
Line  65-75. Animal-Derived Proteins. The author never writes about "white meat" which is an important source of proteins, as well as fish or eggs. A section related to this has been added thank you.
Line 77-97. Plant-derived Proteins. The author could give examples of different plant protein amounts that contributed to human diet.  Results from the NHANEs survey have been added ad an example of this.
Mycelium/Fungal-Based Proteins. Different fungal contained different proteins? Please discuss.  An extra section has been added.
Table 3. Interesting information but the authors could add other data related with plants (for example) I am afraid these are the only examples provided in this reference source that are fully aligned in terms of how they are reported.

Reviewer 2 Report

General comments:

This is a nicely written paper, suggesting that we should consider fungal/mycelium-derived protein as third protein category alongside animal and plant proteins. This suggestion is followed by a review on the composition and research results regarding one particular fungal protein, the mycelium. While the concept of the paper is interesting and timely in light of promoting both sustainable food systems and population health, I find it too one-sided to concentrate only on the mycelium. Its consumption is still rather limited in global scale and to my knowledge, there is only one commercial producer for the mycelium. I think the paper would improve considerably if other major dietary fungal protein sources were introduced and the current state of research regarding them was reviewed, even shortly. That would serve the greater good and be a valuable source of information to those working with Food-Based Dietary Guidelines.

Specific comments:

p.4, from line 150; When describing the observed beneficial effects of mycoprotein, could the author clarify in which dietary settings the effects were seen. For example, was mycoprotein given as a supplement or whole food?

p.5, l. 160; It is too simplistic to assume that all the beneficial metabolic effects of mycoprotein are due to its fibre content, particularly when the fibre seems to be composed mainly of chitin and beta-glucan. The latter may well explain the cholesterol lowering effect of mycoprotein if the ingested dose is big enough. Could the author give an estimate on that?

Otherwise, the bulk of scientific evidence regarding the health effects of dietary fibre is associated with high consumption of cereal fibre, mainly composed of low fermentable fibres accompanied by a variety of bioactive compounds such as polyphenols. Is anything known of possible phytochemicals in mycoprotein? At least some fungi are known to contain polyphenolic compounds.   

Author Response

This is a nicely written paper, suggesting that we should consider fungal/mycelium-derived protein as third protein category alongside animal and plant proteins. This suggestion is followed by a review on the composition and research results regarding one particular fungal protein, the mycelium. While the concept of the paper is interesting and timely in light of promoting both sustainable food systems and population health, I find it too one-sided to concentrate only on the mycelium. Its consumption is still rather limited in global scale and to my knowledge, there is only one commercial producer for the mycelium. I think the paper would improve considerably if other major dietary fungal protein sources were introduced and the current state of research regarding them was reviewed, even shortly. A section on the evidence related to mushrooms has been added. That would serve the greater good and be a valuable source of information to those working with Food-Based Dietary Guidelines.

Specific comments:

p.4, from line 150; When describing the observed beneficial effects of mycoprotein, could the author clarify in which dietary settings the effects were seen. For example, was mycoprotein given as a supplement or whole food?  This has been added thank you.

p.5, l. 160; It is too simplistic to assume that all the beneficial metabolic effects of mycoprotein are due to its fibre content, particularly when the fibre seems to be composed mainly of chitin and beta-glucan The latter may well explain the cholesterol lowering effect of mycoprotein if the ingested dose is big enough. Could the author give an estimate on that?. This has been softened and the phytonutrients mentioned added.  I think a meta-analysis would ideally need to be done in the future to estimate effects.

Otherwise, the bulk of scientific evidence regarding the health effects of dietary fibre is associated with high consumption of cereal fibre, mainly composed of low fermentable fibres accompanied by a variety of bioactive compounds such as polyphenols. Is anything known of possible phytochemicals in mycoprotein? At least some fungi are known to contain polyphenolic compounds.   A section related to this has been added thank you.

Round 2

Reviewer 1 Report

This is an interesting and a very current topic. However, the author should improve the paper, give more information about this thematic, otherwise it's an incipient paper. For example
Line  65-75. Animal-Derived Proteins. The author never writes about "white meat" which is an important source of proteins, as well as fish or eggs.
Line 77-97. Plant-derived Proteins. The author could give examples of different plant protein amounts that contributed to human diet.
Mycelium/Fungal-Based Proteins. Different fungal contained different proteins? Please discuss.
Table 3. Interesting information but the authors could add other data related with plants (for example

Author Response

This is an interesting and a very current topic. However, the author should improve the paper, give more information about this thematic, otherwise it's an incipient paper. For example
Line 65-75. Animal-Derived Proteins. The author never writes about "white meat" which is an important source of proteins, as well as fish or eggs. Three extra paragraphs have been added to the section.  One defining meat in general, including chicken, and a section related to micronutrients and health patterns related to the consumption of the different meat types.  An end paragraph have been added related to animal diseases also which is a very current and thematic issue.
Line 77-97. Plant-derived Proteins. The author could give examples of different plant protein amounts that contributed to human diet. A paragraph relating to the different examples of plant proteins had been added.  Relating this to the diet – a breakdown of plant protein intake and what this was comprised of from the US NHANES survey and a Finnish survey has also been added.
Mycelium/Fungal-Based Proteins. Different fungal contained different proteins? Please discuss. This section has been extended, further divided and sub-sections added to discuss, cover and provide examples of a range of fungal proteins.
Table 3. Interesting information but the authors could add other data related with plants (for example) I am afraid these are the only examples provided in this reference source that are fully aligned in terms of how they are reported.  This Table has now been moved into the introduction as Table 1 as the original section was becoming too long and this subsequently became out of context.  An extra Figure (Figure 1) has also been included in the introduction with should further clarify/answer this point.

Reviewer 2 Report

The author have done some of the changes suggested to the manuscript but I fail to see where the stated ‘section on the evidence related to mushrooms has been added’ is. Is this the text on p. 3, l. 145 to p.4, l. 153? It is not what I meant by saying that the paper in its present form is too one-sided as it only concentrates in the mycelium. That was my major criticism and I suggested that the author would introduce other major dietary fungal/mushroom protein sources and the current state of research regarding them. The rather vague text added is not addressing this. To be in use for drafting Food-Based Dietary Guidelines, the paper needs to cover all fungal protein, not only the mycelium. For example, a table on composition of some most used fungal protein sources globally should be useful to readers and some data on their intakes in different populations as well as an indication of their intakes with any health outcomes. The mycelium can be alongside other fungal protein sources in the text but should not be the only one. What I am looking for is a thorough revision of the text taking into account all fungal protein sources.

Author Response

The author have done some of the changes suggested to the manuscript but I fail to see where the stated ‘section on the evidence related to mushrooms has been added’ is. Is this the text on p. 3, l. 145 to p.4, l. 153? It is not what I meant by saying that the paper in its present form is too one-sided as it only concentrates in the mycelium. That was my major criticism and I suggested that the author would introduce other major dietary fungal/mushroom protein sources and the current state of research regarding them. The rather vague text added is not addressing this. To be in use for drafting Food-Based Dietary Guidelines, the paper needs to cover all fungal protein, not only the mycelium. For example, a table on composition of some most used fungal protein sources globally should be useful to readers and some data on their intakes in different populations as well as an indication of their intakes with any health outcomes. A paragraph relating to fungal protein trends and consumption patterns has been added into the first part of the fungal protein section.  The nutritional composition Table has also been extended to encompass a greater range of fungal proteins.  Specific studies looking at habitual fungal protein intakes in relation to health outcomes are lacking thus difficult to collate.  Intakes have been specified within relevant research studies and review papers that are cited with the main body.   The mycelium can be alongside other fungal protein sources in the text but should not be the only one. What I am looking for is a thorough revision of the text taking into account all fungal protein sources.  The fungal section has been extensively extended and divided into sub-sections to discuss a greater range of fungal protein sources.  A mushroom and other section has been developed.